# Predicting congenital syphilis cases: A performance evaluation of different machine learning models

Igor Vitor Teixeira[1], Morgana Thalita da Silva Leite[1], Flávio Leandro de Morais Melo[1], Élisson da Silva Rocha[1], Sara Sadok[2], Ana Sofia Pessoa da Costa Carrarine[3], Marília Santana[3], Cristina Pinheiro Rodrigues[3], Ana Maria de Lima Oliveira[3], Keduly Vieira Gadelha[3], Cleber Matos de Morais[4], Judith Kelner[5], Patricia Takako Endo[1] *

1 Programa de Pós-Graduação em Engenharia da Computação, Universidade de Pernambuco, Recife, Brazil, 2 Master en Genética Asistencial, Universitat Autònoma de Barcelona, Barcelona, Spain, 3 Secretaria de Saúde do Estado de Pernambuco, Programa Mãe Coruja Pernambucana, Recife, Brazil, 4 Departamento de Mídias Digitais, Universidade Federal da Paraíba, João Pessoa, Brazil, 5 Centro de Informática, Universidade Federal de Pernambuco, Recife, Brazil

* patricia.endo@upe.br

**Data Availability Statement:** All data are available from the Mendeley Data database (accession number(s) doi:10.17632/3zkcvybvkz.1).

## Abstract

### Background

Communicable diseases represent a huge economic burden for healthcare systems and for society. Sexually transmitted infections (STIs) are a concerning issue, especially in developing and underdeveloped countries, in which environmental factors and other determinants of health play a role in contributing to its fast spread. In light of this situation, machine learning techniques have been explored to assess the incidence of syphilis and contribute to the epidemiological surveillance in this scenario.

### Objective

The main goal of this work is to evaluate the performance of different machine learning models on predicting undesirable outcomes of congenital syphilis in order to assist resources allocation and optimize the healthcare actions, especially in a constrained health environment.

### Method

We use clinical and sociodemographic data from pregnant women that were assisted by a social program in Pernambuco, Brazil, named *Mãe Coruja Pernambucana* Program (PMCP). Based on a rigorous methodology, we propose six experiments using three feature selection techniques to select the most relevant attributes, pre-process and clean the data, apply hyperparameter optimization to tune the machine learning models, and train and test models to have a fair evaluation and discussion.

**Funding:** This work was supported, in whole or in part, by the Bill & Melinda Gates Foundation [OPP1202194]. Under the grant conditions of the Foundation, a Creative Commons Attribution 4.0 Generic License has already been assigned to the Author Accepted Manuscript version that might arise from this submission. There was no additional external funding received for this study.

**Competing interests:** The authors have declared that no competing interests exist.

**Abbreviations:** AdaBoost, Adaptive Boosting; AWS, Amazon Web Services; BDS, Balanced Data Set; BODDS, Balanced with One-hot Encoding with Column Drop Data Set; BODS, Balanced with One-hot Encoding Data Set; CAAE, Certificate of Presentation of Ethical Appreciation; CEP, *Comitê de Ética em Pesquisa*; EC2, Amazon Elastic Compute Cloud; FN, False Negative; FP, False Positive; GBM, Gradient Boosting Machines; ICD-10, International Classification of Diseases 10th Revision; IDS, Imbalanced Data Set; IODDS, Imbalanced with One-hot Encoding with Column Drop Data Set; IODS, Imbalanced with One-hot Encoding Data Set; KNN, K-Nearest Neighbors; PAHO, Pan American Health Organization; PHC, Primary Health Care; PMCP, *Mãe Coruja Pernambucana*; SBS, Sequential Backward Selection; SFA, Sequential Feature Algorithm; SFS, Sequential Forward Selection; SIH, *Sistema de Informações Hospitalares*; SIM, *Sistema de Informação Sobre Mortalidade*; SINAN, *Sistema de Informação de Agravos de Notificação*; SINASC, *Sistema de Informações sobre Nascidos Vivos*; SIS-MC, *Sistema de Informação do Mãe Coruja*; STIs, Sexually Transmitted Infections; SUS, *Sistema Único de Saúde*; SVM, Support Vector Machine; TN, True Negative; TP, True Positive; UFPE, *Universidade Federal de Pernambuco*; VDRL, Venereal Disease Research Laboratory; WHO, World Health Organization; XGBoost, eXtreme Gradient Boosting.

## Results

The AdaBoost-BODS-Expert model, an Adaptive Boosting (AdaBoost) model that used attributes selected by health experts, presented the best results in terms of evaluation metrics and acceptance by health experts from PMCP. By using this model, the results are more reliable and allows adoption on a daily usage to classify possible outcomes of congenital syphilis using clinical and sociodemographic data.

## Introduction

With major unfavorable outcomes and high economic costs, (STIs) are a structural problem in global health. According to the (WHO), more than one million STIs are contracted daily, and an estimated 374 million new infections of chlamydia, gonorrhea, trichomoniasis, and syphilis occur annually around the world [1].

Caused by viruses, bacteria, and other microorganisms, STIs can be spread through sexual contact, by sharing non-sterile sharp objects, through transfusion of infected blood, or congenitally. Currently, there are more than 30 pathological agents that can cause STIs [1]. The impact in the healthcare systems of these disorders are especially significant in countries with large populations living in poverty [2–4].

Moreover, underdeveloped and developing countries are more susceptible to epidemics due to environmental factors such as lack of resources for healthcare facilities, low development stage, and lack of information on contraceptive methods [1]. Therefore, the epidemiological scenario of STIs is worrisome, especially for syphilis, which can generate serious complications and even death in infected individuals who were not diagnosed or not received adequate treatment. The latest survey carried out by WHO [1] presented an annual forecast of 7.1 million new infections of syphilis around the world. Gestational syphilis is the second leading cause of stillbirths worldwide and also results in premature birth, low birth weight, neonatal death, and physical deformities in newborns, among other conditions [5].

Syphilis is an infection caused by the bacterium *Treponema pallidum*, and presents with a variety of symptoms according to the different stages of the disorder (primary, secondary, and tertiary). The highest risk of contagion is in the primary and secondary stages [6]. Even so, syphilis can be treated and is easily curable. However, the different manifestations of the disorder, such as multi-system involvement in chronic infections, gestational syphilis and congenital syphilis have a great impact on global public health, as they have adverse aggravating results for pregnant women, pregnancies, and children [5].

According to the Syphilis Epidemiological Bulletin 2021 of the Ministry of Health of Brazil [7], between 2011 and 2020, it was registered an unbridled growth of the incidence rate of congenital syphilis, rising from 3.3 cases/1,000 live births to 7.7 cases, reaching a peak in 2018 with the incidence rate of 9 cases/1,000 live births. In 2020, 20,065 cases of congenital syphilis were recorded and an infant mortality coefficient of 6.5/100,000 live births with 186 deaths due to congenital syphilis were reported in Brazil.

In 2020, complying actively with the national epidemiological scenario, according to the bulletin [7], the state of Pernambuco presented an incidence rate of 13.1 cases of congenital syphilis in children under one year per 1,000 live births, 26 times greater than the WHO and Pan American Health Organization (PAHO) recommendations [8], the 3rd highest rate in Brazil, higher than the national average rate.

Researchers, humanitarian and governmental institutions seek through research projects and public policies to minimize and eradicate the incidence of syphilis in Primary Health Care (PHC), in particular, congenital syphilis. The *Mãe Coruja Pernambucana* Program (PMCP) [9] is a great example of a social program created by the government of the state of Pernambuco, present in more than 105 vulnerable municipalities in the state of Pernambuco, that aims to offer support to pregnant women through *Sistema Único de Súude* (SUS), the Brazilian public healthcare system, before and after the birth of their children, monitoring the evolution of children up to the age of 5 years, guaranteeing them a healthy and harmonious development during the first years of life.

The rapid test for syphilis is available free of charge to the entire Brazilian population [10]. It has been used as a fundamental tool for the increase of the rate of gestational syphilis detection. Its scarcity in the PHC, however, is common and can contribute to masking of the real number of positive cases and, consequently, underestimate the number of people who should be under treatment and follow-up [11].

The detection of the main clinical and sociodemographic factors related to congenital syphilis has proved to be a challenge. In addition, the scarcity of resources for public health in Brazil [12] has highlighted the need for innovative solutions, such as the use of machine learning techniques. Machine learning techniques can help the decision making process through analysis of the clinical and sociodemographic data of the pregnant women assisted by the PMCP, helping health professionals for better monitoring of pregnant women. After being trained, computational models have a low operating cost, facilitating the implementation and the usage of these technological tools in low-resources communities.

Some works in the literature have proposed machine learning models to predict the incidence of syphilis cases in a population using data from Twitter [13], and profiling syphilis patients [14]. Differently from the current literature, this work aims to present a performance evaluation of different machine learning models to classify possible undesirable outcomes of congenital syphilis, using data from pregnant women assisted by the PMCP. The model uses clinical and sociodemographic data as input, enabling better monitoring and care during gestation.

## Related works

Some works in the literature have already presented studies on the incidence and the risk factors for syphilis, such as Santos et al. [11] that presented a study to understand the factors related to the trends of syphilis in Brazil, analyzing their association with sociodemographic aspects and the PHC in the period between 2011 and 2019. Authors concluded that there are some important predictors of the upward trends of acquired syphilis, such as the quality of the PHC service, the availability of penicillin in the PHC, the availability of the female condom, and the influence of population size. The need for training of the health professionals in the care of STIs is also reinforced as an indispensable action to control syphilis cases.

Lima et al. [15] described the incidence of congenital syphilis in the city of Belo Horizonte, capital of Minas Gerais a Brazilian state, between the years 2001 and 2008, aiming to identify the risk factors associated with the diagnosis of the disease. A multivariate logistic regression analysis was performed to identify maternal and antenatal characteristics independently associated with the occurrence of congenital syphilis. It was identified as an independent risk factor for congenital syphilis characteristics such as black-skinned mothers, maternal educational level, and the absence of antenatal care, which is the main risk factor, presenting an 11 times greater chance of congenital syphilis than pregnancies that the mother attended at least one time to the antenatal follow-up.

Melo et al. [16] analyzed the association between morbidity from congenital syphilis and biological, socioeconomic, and antenatal care indicators using data from the city of Recife, capital of the state of Pernambuco, between 2004 and 2006. According to their results, factors such as fewer than four antenatal appointments, mothers under 20 years of age, and black and mixed skin color were associated with cases of congenital syphilis. The increase in risk was proportional to the deterioration of socioeconomic and biological indicators and antenatal care, and the worst situation occurred in more peripheral areas of the cities.

Young et al. [13] conducted a study to explore whether data from a social network (Twitter) could be used to identify trends in cases of syphilis, primary and secondary, and late latent syphilis in the United States of America. They sought to identify associations between tweets related to sexual risks, that would possibly be reported in the following year. For the years 2012 and 2013, weekly disease data from counties and the capital were collected. According to the results, counties and capitals with the highest number of risk-related tweets in 2012 were associated with a 2.7% increase in primary and secondary syphilis cases and a 3.6% increase in latent syphilis cases in 2013. Their results were consistent in all models run in the analysis, suggesting a relationship between syphilis and risk-related tweets.

Silva [14] developed a research close to the objective of our work and, therefore, could be used for later comparison. A predictive analysis was performed to create a profile of risk groups using historical data generated by systems used by SUS, *Sistema de Informaçãao Sobre Mortalidade* (SIM), the Brazilian death registration system, *Sistema de Informação de Agravos de Notificação* (SINAN), and *Sistema de Informaçõoes Hospitalares* (SIH), the Brazilian hospital admissions system. The data related to cases of acquired, gestational, and congenital syphilis in Brazil between 2010 and 2019 were used. Due to difficulties in linking all the syphilis-related attributes available in the databases, only data related to sex, color, educational level, and age group were selected. After applying clustering techniques, 130 profiles were found, obtaining an accuracy of 97.41%.

Our work differs from others in the literature by conducting and discussing a performance evaluation of machine learning models using clinical and sociodemographic data presented in an integrated system that records data from antenatal care, birth, and child's development monitoring. This data allows us to focus on the classification of possible cases of congenital syphilis in a full range of relationships and evaluate which machine learning models are more efficient in our scenario.

## Background

**Machine learning models.** In this work, we evaluate the following machine learning models for the classification of congenital syphilis cases: Decision Tree, Random Forest, Adaptive Boosting (AdaBoost), Gradient Boosting Machines (GBM), eXtreme Gradient Boosting (XGBoost), K-Nearest Neighbors (KNN), and Support Vector Machine SVM.

A Decision Tree [17, 18] is a supervised machine learning algorithm that may be used to classify categorical values classification tree or predict numerical values regression tree by making a statement followed by a decision depending on whether the statement is true or false. Basically, a Decision Tree consists of a root node, internal nodes, and leaf nodes, which are created by gradually splitting the data based on the target attribute. A root node is identified by the attribute that best splits the data, focusing on the target attribute. In other words, the attribute with the lowest impurity value. Then, to reduce the impurity, the root node is split, creating the internal nodes. The procedure of splitting the nodes is performed until it is no longer possible to reduce the impurity value, resulting in the formation of leaf nodes, which indicate the outputs of the Decision Tree.

Random Forest [19] consists of an ensemble of Decision Trees, each of which is trained by a subset of random attributes extracted from the training data set, a technique known as bootstrap. The results of each Decision Tree are then aggregated according to the classification of the class, and the class with the most votes is chosen by majority vote.

The AdaBoost is based on the idea of boosting, in machine learning this means creating a highly accurate prediction by combining many weak predictors [20]. In the algorithm, given the training data set, a distribution is calculated over the examples using a weak learner with the objective of finding a classifier with a low error relative to the distribution. This process is repeated *n* times, and the final classifier is a weighted combination of the classifications of weak learners [21].

GBM [22] does a sequential procedure to predict values based on previous errors. Gradient refers to the error gained after building a model, and boosting relates to improvement. In the GBM algorithm, the predictions are started by a simple Decision Tree. The residual is calculated by subtracting the actual value from the predicted value. Another shallow Decision Tree is built that predicts residuals based on all the independent values. Then, the original prediction is updated with the new prediction multiplied by the learning rate. The last three steps are repeated for a certain number of iterations, known as the number of trees.

As its core, XGBoost has a Decision Tree boosting algorithm that attempts to correct or minimize the errors in the previous model. The XGBoost algorithm is based on a generalized GBM, but it also includes a regularization term to prevent over-fitting and supports arbitrary differentiable loss functions [23].

KNN is an algorithm that selects the *k* closest samples, or *k* nearest neighbors, in the training data set based on a distance metric such as Euclidean distance, and then predicts the class based on the major class from those samples in the *k* nearest neighbors [24].

SVM aims to find linearity between data by using hyperplanes, even if the data is not linearly separable [25]. For that, SVM uses the kernel concept to map the scenario data initially presented by the variables to a high-dimensional space to enhance separation between classes [26]. Due to specific aspects of understanding how to manipulate non-linear data, SVM has become a very popular model in the healthcare scenario [27].

**Data balancing and encoding techniques.** The purpose of data balancing is to equalize the amount of data of the target attribute whose positive and negative classes are out of balance. According to [28, 29], data imbalance is one of the obstacles that hinder the learning of classification algorithms, as it can lead to a learning bias in which the model learns more about the majority class than the minority, resulting in models with low performance due to the disparity between classes. One of the ways to get around the problem is the random under-sampling technique, which [30] presents as a heuristic method that randomly eliminates instances of the majority class until the quantity is balanced with the minority class.

Categorical classes are typically encoded to binary or numeric values using computational coding techniques prior to training machine learning models. According to [31], one of the most widely used encoding techniques used in the literature is the one-hot encoding technique, which seeks to transform each categorical attribute into new attributes with binary values where a value of one indicates the presence and a value of zero indicates the absence of the coded categorical value specific for the new attribute [32].

**Hyper-parameter optimization and feature selection techniques.** Machine learning models might have several parameters, also known as hyper-parameters, to configure, making manual configuration impractical. Automated hyper-parameter optimization techniques, such as Grid Search, were used in this work to determine the best hyper-parameter combination for each model. The Grid Search technique [33] performs an exhaustive search for training and

evaluating models with all the combinations of hyper-parameters in a given search space, returning the hyper-parameters that achieve the best performance.

As is the case with this work, data sets with numerous attributes (high dimensionality) may result in the preponderance of noisy, irrelevant, and redundant data [34], thus impacting the machine learning models' learning. To deal with it, feature selection techniques could be used to reduce the dimensionality of the data set by selecting a relevant subset of attributes from the original data set. These techniques can be categorized into three approaches: filter, wrapper, and embedded [35]. In this research, we used a wrapper approach.

Computationally more expensive, the wrapper approach uses a machine learning model to select and evaluate the subset of attributes with the best performance, running until a stop condition is satisfied [34]. For this work, we set the stop condition equal to the number of attributes of the data set, forcing the technique to evaluate all possible subsets of attributes and select the one with the highest performance. We used the Sequential Feature Algorithm (SFA) technique to implement the wrapper approach, focusing on Sequential Forward Selection (SFS) and Sequential Backward Selection (SBS), which are its two main types [36].

The SFS flavor starts with an empty subset of attributes; with each iteration, a new attribute is added, thereby selecting the attributes that increase the model's performance. On the other hand, the SBS flavor begins with a subset including all the attributes of the data set; with each iteration, different combinations of attributes are evaluated, and the attribute with the least impact on model performance is removed.

**Evaluation metrics.** In this research, we evaluated the proposed models using the accuracy, precision, sensitivity, specificity, and F1-Score metrics. All these evaluation metrics are based on the Confusion Matrix [37], which seeks to calculate: (*i*) True Positive (TP) for classified as positive and really positive, (*ii*) False Positive (FP) for classified as positive but actually negative, (*iii*) True Negative (TN) for classified as negative and really negative, and (*iv*) False Negative (FN) for classified as negative but actually positive.

Accuracy [38] measures the model's performance according to the total samples correctly classified, indicating how frequently the model was correct, and is defined as

$$\text{accuracy} = \frac{\text{TP} + \text{TN}}{\text{TP} + \text{TN} + \text{FP} + \text{FN}} \tag{1}$$

Precision is a metric used to determine the proportion of positive classifications that are actually positive in reality [39], calculated as

$$\text{precision} = \frac{\text{TP}}{\text{TP} + \text{FP}} \tag{2}$$

Sensitivity determines the proportion of real positives that were correctly classified [40], defined as

$$\text{sensitivity} = \frac{\text{TP}}{\text{TP} + \text{FN}} \tag{3}$$

Specificity is the inverse of sensitivity, which seeks to determine the proportion of real negatives that were correctly classified [40], calculated as

$$\text{specificity} = \frac{\text{TN}}{\text{TN} + \text{FP}} \tag{4}$$

F1-Score [41] is a metric that calculates the harmonic mean of precision and sensitivity to summarize the predictive performance of the models, and is defined as

$$\text{F1} - \text{Score} = 2 \times \frac{precision \times recall}{precision + recall} \tag{5}$$

# Materials and methods

## Ethics approval and consent to participate

We declare that the research has been approved by the Brazilian Human Research Ethics Board (*Comitê de Ética em Pesquisa* [CEP]) under number 12438019.2.0000.5208 and all methods were performed in accordance with the Brazilian regulations that do not require consent for studies using unidentified data from the Brazilian data health systems.

## Data set

We used nine tables with anonymized data provided by the PMCP, extracted from their information system, named *Sistema de Informação do Mãae Coruja* (SIS-MC). These tables contain clinical and sociodemographic data regarding antenatal care, pregnant women's outcomes, and their children, from the cities served by the PMCP in the State of Pernambuco, Brazil, between the years of 2013 and 2021. The pre-processed data set is publicly available at https://data.mendeley.com/datasets/3zkcvybvkz/1. The use of data from the (SIS-MC) was authorized by the *Comitê deÉtica em Pesquisa* (CEP) of the *Universidade Federal de Pernambuco* (UFPE) with the (CAAE) number 12438019.2.0000.5208 and authorized by the partner institution, the Certificate of Presentation of Ethical Appreciation (PMCP).

Fig 1 illustrates the data pre-processing methodology designed to unify the nine tables provided by the PMCP into an unified data set and to perform manual feature selection, handle missing data, remove outliers, and create new attributes.

A unification step was applied as the data provided by the PMCP were shared across nine tables. At first, we selected tables that included any clinical and sociodemographic data regarding antenatal care, pregnant women's outcomes, and their children. Fig 2 presents the nine selected tables. Since the goal of this work is to predict undesirable outcomes of congenital syphilis, the children's table was chosen as the starting point for the unification because it includes the result of the Venereal Disease Research Laboratory (VDRL) test, which is a screening test for congenital syphilis at birth and was chosen as the target attribute of the classification.

The target attribute had 807 positive cases of congenital syphilis, 40,995 negative cases, and 162,741 empty records. We used the tables related to abnormal findings and morbidities related to children's table to find other possible cases of congenital syphilis not recorded by the target attribute. The abnormal findings and morbidities selected are presented in Table 1 with their respective International Classification of Diseases 10th Revision (ICD-10). Therefore, only 17 records with empty VDRL test result and 3 negative cases were changed to positive. This resulted in a target attribute with 827 cases of congenital syphilis positive, 40,992 cases negative, and 162,724 empty records, which were removed from the data set.

The children's table was unified with tables related to childbirth, pregnancies, pregnant women, and antenatal care. At the final stage of data unification, the unified data set contained 204,543 records and 218 attributes.

We conducted a manual feature selection in order to select relevant clinical and sociodemographic attributes related to pregnancies, pregnancy outcomes, and pregnant women, and reduce the data dimensionality. The attribute with the result of the VDRL test, which is a

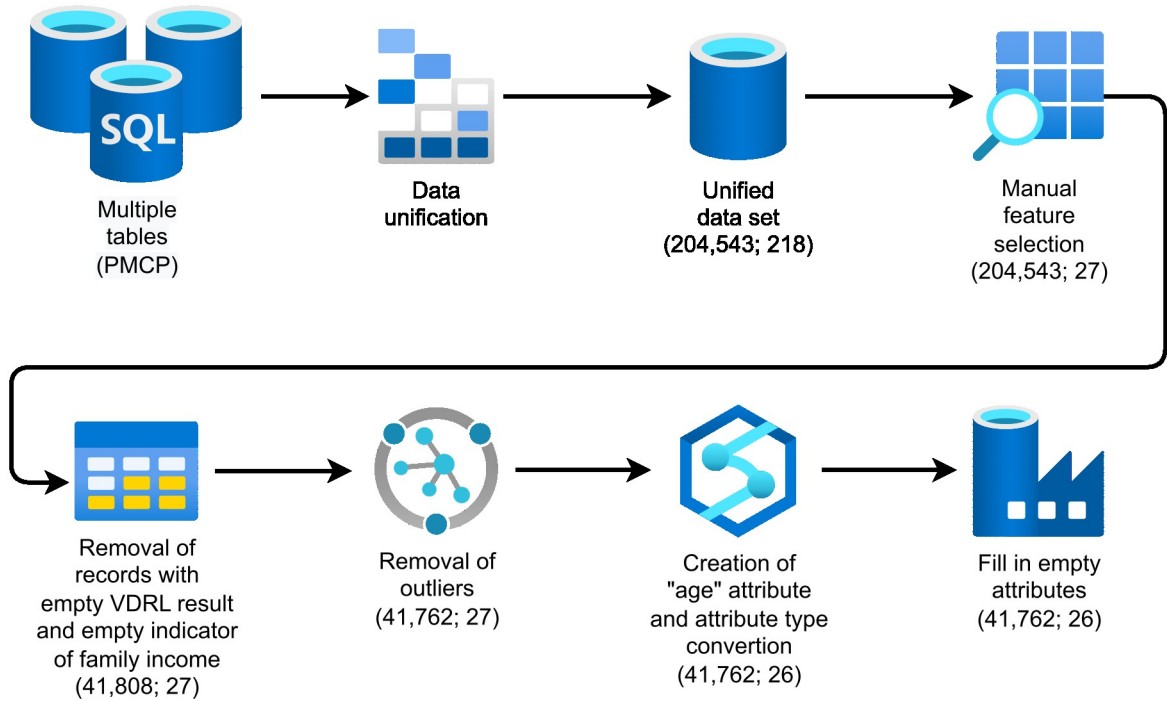

**Fig 1. Data pre-processing methodology.**

screening test for congenital syphilis at childbirth, was chosen as our target attribute. To clean up the data set, we removed attributes with more than 70% missing data, except for the target attribute, reducing the dimensionality to 27 attributes.

The target attribute has 827 positive cases of congenital syphilis, 40,992 negative cases, and 162,724 empty records (that were removed from the data set). 11 records were removed with an empty value for the family income indicator, as informed in the pregnant woman's record in SIS-MC, which resulted in a data set with 41,808 records. These records were related to negative cases of congenital syphilis, and as the data set has numerous negative cases, especially when compared to the number of positive cases, they were removed without negatively impacting on data quality.

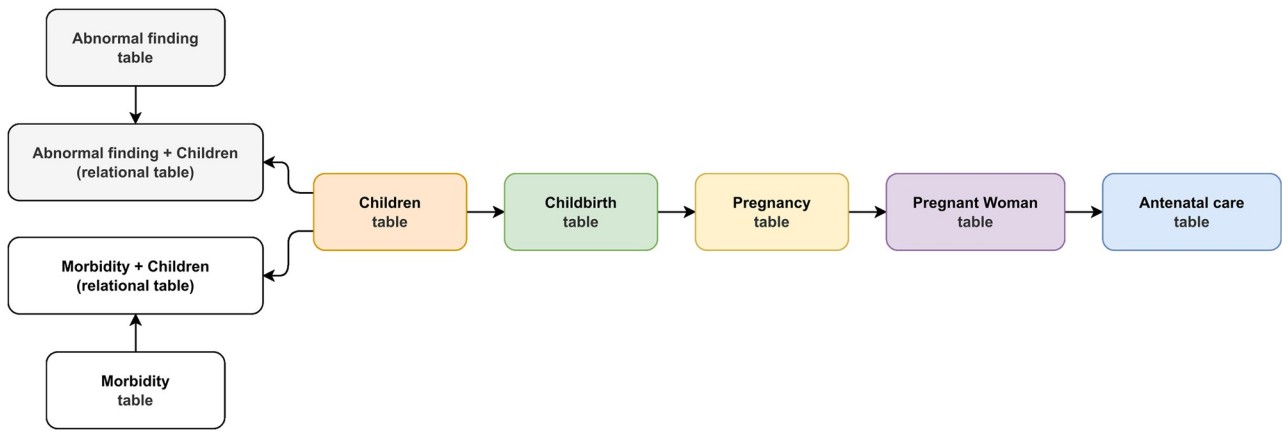

**Fig 2. Original tables mapping.**

**Table 1. Abnormal findings and morbities related to syphilis.**

| Data set | ICD-10 | Description |
|---|---|---|
| Abnormal findings | O98 | Maternal infectious and parasitic diseases classifiable elsewhere but complicating pregnancy, childbirth, and the puerperium |
| Morbidity | A50.0 | Early congenital syphilis, symptomatic |
| Morbidity | A50.1 | Early congenital syphilis, latent |
| Morbidity | A50.2 | Early congenital syphilis, unspecified |
| Morbidity | A50.4 | Late congenital neurosyphilis [juvenile neurosyphilis] |
| Morbidity | A50.5 | Other late congenital syphilis, symptomatic |
| Morbidity | A50.6 | Late congenital syphilis, latent |
| Morbidity | A50.7 | Late congenital syphilis, unspecified |
| Morbidity | A50.9 | Congenital syphilis, unspecified |
| Morbidity | A51.0 | Primary genital syphilis |
| Morbidity | A51.1 | Primary anal syphilis |
| Morbidity | A51.2 | Primary syphilis of other sites |
| Morbidity | A51.3 | Secondary syphilis of skin and mucous membranes |
| Morbidity | A51.4 | Other secondary syphilis |
| Morbidity | A51.5 | Early syphilis, latent |
| Morbidity | A51.9 | Early syphilis, unspecified |
| Morbidity | A52.0 | Cardiovascular and cerebrovascular syphilis |
| Morbidity | A52.1 | Symptomatic neurosyphilis |
| Morbidity | A52.2 | Asymptomatic neurosyphilis |
| Morbidity | A52.3 | Neurosyphilis, unspecified |
| Morbidity | A52.7 | Other symptomatic late syphilis |
| Morbidity | A52.8 | Late syphilis, latent |
| Morbidity | A52.9 | Late syphilis, unspecified |
| Morbidity | A53.0 | Latent syphilis, unspecified as early or late |
| Morbidity | A53.9 | Syphilis, unspecified |
| Morbidity | A65 | Nonvenereal syphilis |

After analyzing the baseline characteristics of the data set with the help of health experts (stakeholders) from the (PMCP), some records considered as outliers were also removed, referring to *(i)* pregnant women who were born before 1960 and after 2020; *(ii)* family income informed at antenatal care greater than 20,000; and *(iii)* number of residents in the household greater than 20.

The age of the pregnant women when they were attended by the PMCP attribute was created, and to calculate it, we subtracted the date of registration of the pregnancy in the SIS-MC from the date of birth of the pregnant woman. After that, these two attributes (date of registration and date of birth) were removed from the data set. We also converted attributes from numerical to categorical, such as the number of abortions, number of living children, number of pregnancies, and number of residents in the household. At the end, the data set was left with 41,762 records and 26 attributes.

At last, we created a new category to fill in the empty data. We use this strategy because all our attributes were binary or categorical (except for the age attribute), as can be seen in Table 2, which presents the data set attributes after all modifications.

After all pre-processing steps, the pre-processed data set contained 826 positive cases and 40,936 negative cases of congenital syphilis. The overall baseline characteristics of the pre-processed data set related to the pregnant women assisted by the PMCP are presented in Table 3.

**Table 2. Data set attributes.**

| Attribute | Description | Type | Categorization |
|---|---|---|---|
| VDRL_RESULT▲ | VDRL result | Binary | (i) Positive and (ii) Negative |
| CONS_ALCOHOL | Consume alcohol | Categorical | (i) Positive, (ii) Negative, and (iii) Not informed |
| RH_FACTOR | RH factor | Categorical | (i) Positive, (ii) Negative, and (iii) Not informed |
| SMOKER | Smoker | Categorical | (i) Positive, (ii) Negative, and (iii) Not informed |
| PLAN_PREGNANCY★ | Planned pregnancy | Categorical | (i) Positive, (ii) Negative, and (iii) Not informed |
| BLOOD_GROUP | Blood group | Categorical | (i) Positive, (ii) Negative, and (iii) Not informed |
| HAS_PREG_RISK★ | Has pregnancy risk | Categorical | (i) O, (ii) A, (iii) B, (iv) AB, and (v) Not informed |
| TET_VACCINE | Tetanus vaccination | Categorical | (i) Positive, (ii) Negative, and (iii) Not informed |
| IS_HEAD_FAMILY | Is head of family | Categorical | (i) Positive, (ii) Negative, and (iii) Not informed |
| MARITAL_STATUS★ | Marital status | Categorical | (i) Single, (ii) Married, (iii) Widowed, (iv) Judicial separation, (v) Divorced, and (vi) Not informed |
| FOOD_INSECURITY★ | Food insecurity | Categorical | (i) Positive, (ii) Negative, and (iii) Not informed |
| NUM_ABORTIONS★ | Number of abortions | Categorical | (i) None, (ii) One, (iii) Two, (iv) More than two, and (v) Not informed |
| NUM_LIV_CHILDREN★ | Number of living children | Categorical | (i) None, (ii) One, (iii) Two, (iv) More than two, and (v) Not informed |
| NUM_PREGNANCIES★ | Number of pregnancies | Categorical | (i) None, (ii) One, (iii) Two, (iv) More than two, and (v) Not informed |
| FAM_PLANNING★ | Received information about family planning | Categorical | (i) Positive, (ii) Negative, and (iii) Not informed |
| TYPE_HOUSE | Type of house construction | Categorical | (i) Straw, (ii) Wood, (iii) Clay, (iv) Plaster, (v) Masonry, and (vi) Not informed |
| HAS_FAM_INCOME | Has family income | Binary | (i) Positive and (ii) Negative |
| EDUC_LEVEL★ | Educational level | Categorical | (i) Complete elementary school, (ii) Incomplete elementary school, (iii) Complete middle school, (iv) Incomplete middle school, (v) Complete high school, (vi) Incomplete high school, (vii) Complete superior school, (viii) Incomplete superior school, and (ix) Not informed |
| CONN_SEWER_NET | House connected to the sewer network | Categorical | (i) Positive, (ii) Negative, and (iii) Not informed |
| NUM_RES_HOUSEHOLD | Number of residents in the household | Categorical | (i) None, (ii) One, (iii) Two, (iv) Three, (v) More than three, and (vi) Not informed |
| HAS_FRU_TREE | Has fruit trees | Categorical | (i) Positive, (ii) Negative, and (iii) Not informed |
| HAS_VEG_GARDEN | Has a vegetable garden | Categorical | (i) Positive, (ii) Negative, and (iii) Not informed |
| FAM_INCOME★ | Family income informed at antenatal cares | Categorical | (i) Less than 500, (ii) Between 501 and 1000, (iii) More than 1000, and (iv) Not informed |
| HOUSING_STATUS | Housing status | Categorical | (i) Owned, (ii) Rented, (iii) Donated, and (iv) Not informed |
| WATER_TREATMENT | Type of water treatment used | Categorical | (i) Filtered, (ii) Boiled, (iii) Disinfected (chlorine), (iv) None, and (v) Not informed |
| AGE★ | Age of the pregnant woman | Numerical | – |

▲ Target attribute for classification;

★ Attribute manually selected by health experts from PMCP

## Experiments' methodology

In this work, we defined six different experiments in order to compare the performance of machine learning models when handling different configurations of data sets. The main idea is to compare the impacts of using *(i)* imbalanced data and *(ii)* the one-hot encoding technique on the models' learning and performance. We evaluate the following machine learning techniques: Decision Tree, Random Forest, AdaBoost, GBM, XGBoost, KNN, and SVM. For each experiment, we built a data set according to the following characteristics:

1. **Imbalanced Data Set (IDS)**: imbalanced data set with 2,327 records (826 positive cases and 1,501 negative cases) and 26 attributes. As the original data set contained numerous negative cases when compared to the number of positive cases (40,936 negative cases and 826

**Table 3. Baseline characteristics of the data set.**

| Variables | Total | Positive | Negative |
|---|---|---|---|
| Total: n (%) | 41,762 | 826 | 40,936 |
| Consume alcohol: n (%) | | | |
| Yes | 1,263 (3.0) | 60 (7.3) | 1,203 (2.9) |
| No | 36,359 (87.1) | 684 (82.8) | 35,675 (87.1) |
| Missing | 4,140 (9.9) | 82 (9.9) | 4,058 (9.9) |
| RH Factor: n (%) | | | |
| Positive | 25,761 (61.7) | 448 (54.2) | 25,313 (61.8) |
| Negative | 2,151 (5.2) | 37 (4.5) | 2,114 (5.2) |
| Missing | 13,850 (33.2) | 341 (41.3) | 13,509 (33.0) |
| Smoker: n (%) | | | |
| Yes | 1,479 (3.5) | 78 (9.4) | 1,401 (3.4) |
| No | 37,105 (88.8) | 684 (82.8) | 36,421 (89.0) |
| Missing | 3,178 (7.6) | 64 (7.7) | 3,114 (7.6) |
| Planned pregnancy: n (%) | | | |
| Yes | 16,772 (40.2) | 274 (33.2) | 16,498 (40.3) |
| No | 22,889 (54.8) | 521 (63.1) | 22,368 (54.6) |
| Missing | 2,101 (5.0) | 31 (3.8) | 2,070 (5.1) |
| Blood group: n (%) | | | |
| O | 13,100 (31.4) | 234 (28.3) | 12,866 (31.4) |
| A | 10,350 (24.8) | 169 (20.5) | 10,181 (24.9) |
| B | 3,457 (8.3) | 56 (6.8) | 3,401 (8.3) |
| AB | 1,075 (2.6) | 28 (3.4) | 1,047 (2.6) |
| Missing | 13,780 (33.0) | 339 (41.0) | 13,441 (32.8) |
| Has pregnancy risk: n (%) | | | |
| Yes | 5,406 (12.9) | 145 (17.6) | 5,261 (12.9) |
| No | 34,362 (82.3) | 636 (77.0) | 33,726 (82.4) |
| Missing | 1,994 (4.8) | 45 (5.4) | 1,949 (4.8) |
| Got tetanus vaccine: n (%) | | | |
| Yes | 36,726 (87.9) | 743 (90.0) | 35,983 (87.9) |
| No | 3,185 (7.6) | 69 (8.4) | 3,116 (7.6) |
| Missing | 1,851 (4.4) | 14 (1.7) | 1,837 (4.5) |
| Head of family: n (%) | | | |
| Yes | 6,074 (14.5) | 154 (18.6) | 5,920 (14.5) |
| No | 31,274 (74.9) | 581 (70.3) | 30,693 (75.0) |
| Missing | 4,414 (10.6) | 91 (11.0) | 4,323 (10.6) |
| Marital status: n (%) | | | |
| Single | 13,658 (32.7) | 277 (33.5) | 13,381 (32.7) |
| Married | 10,102 (24.2) | 115 (13.9) | 9,987 (24.4) |
| Widowed | 75 (0.2) | 1 (0.1) | 74 (0.2) |
| Judicial separation | 93 (0.2) | 4 (0.5) | 89 (0.2) |
| Divorced | 280 (0.7) | 8 (1.0) | 272 (0.7) |
| Other | 17,554 (42.0) | 421 (51.0) | 17,133 (41.9) |
| Food insecurity: n (%) | | | |
| Yes | 7,129 (17.1) | 135 (16.3) | 6,994 (17.1) |
| No | 16,487 (39.5) | 234 (28.3) | 16,253 (39.7) |
| Missing | 18,146 (43.5) | 457 (55.3) | 17,689 (43.2) |
| Number of abortions: n (%) | | | |

*(Continued)*

**Table 3.** (Continued)

| Variables | Total | Positive | Negative |
|---|---|---|---|
| None | 13,421 (32.1) | 231 (28.0) | 13,190 (32.2) |
| One | 4,962 (11.9) | 126 (15.3) | 4,836 (11.8) |
| More than one | 1,333 (3.2) | 28 (3.4) | 1,305 (3.2) |
| Missing | 22,046 (52.8) | 441 (53.4) | 21,605 (52.8) |
| Number of living children: n (%) | | | |
| None | 4,559 (10.9) | 80 (9.7) | 4,479 (10.9) |
| One | 9,685 (23.2) | 187 (22.6) | 9,498 (23.2) |
| Two | 4,817 (11.5) | 113 (13.7) | 4,704 (11.5) |
| More than two | 3,583 (8.6) | 87 (10.5) | 3,496 (8.5) |
| Missing | 19,118 (45.8) | 359 (43.5) | 18,759 (45.8) |
| Number of pregnancies: n (%) | | | |
| None | 4,477 (10.7) | 86 (10.4) | 4,391 (10.7) |
| One | 11,533 (27.6) | 196 (23.7) | 11,337 (27.7) |
| Two | 8,135 (19.5) | 173 (20.9) | 7,962 (19.4) |
| More than two | 9,046 (21.7) | 201 (24.3) | 8,845 (21.6) |
| Missing | 8,571 (20.5) | 170 (20.6) | 8,401 (20.5) |
| Received information about family planning: n (%) | | | |
| Yes | 19,831 (47.5) | 338 (40.9) | 19,493 (47.6) |
| No | 12,005 (28.7) | 265 (32.1) | 11,740 (28.7) |
| Missing | 9,926 (23.8) | 223 (27.0) | 9,703 (23.7) |
| House construction: n (%) | | | |
| Straw | 150 (0.4) | 2 (0.2) | 148 (0.4) |
| Wood | 137 (0.3) | 6 (0.7) | 131 (0.3) |
| Clay | 699 (1.7) | 14 (1.7) | 685 (1.7) |
| Plaster | 70 (0.2) | 5 (0.6) | 65 (0.2) |
| Masonry | 37,929 (90.8) | 738 (89.3) | 37,191 (90.9) |
| Missing | 2,777 (6.6) | 61 (7.4) | 2,716 (6.6) |
| Has family income: n (%) | | | |
| Yes | 31,261 (74.9) | 609 (73.7) | 30,652 (74.9) |
| No | 10,501 (25.1) | 217 (26.3) | 10,284 (25.1) |
| Level of schooling: n (%) | | | |
| Illiterate | 527 (1.3) | 19 (2.3) | 508 (1.2) |
| Complete elementary school | 1,470 (3.5) | 35 (4.2) | 1,435 (3.5) |
| Incomplete elementary school | 5,131 (12.3) | 144 (17.4) | 4,987 (12.2) |
| Complete middle school | 2,521 (6.0) | 49 (5.9) | 2,472 (6.0) |
| Incomplete middle school | 9,218 (22.1) | 214 (25.9) | 9,004 (22.0) |
| Complete high school | 13,166 (31.5) | 185 (22.4) | 12,981 (31.7) |
| Incomplete high school | 6,777 (16.2) | 150 (18.2) | 6,627 (16.2) |
| Complete superior school | 1,063 (2.5) | 7 (0.8) | 1,056 (2.6) |
| Incomplete superior school | 847 (2.0) | 7 (0.8) | 840 (2.1) |
| Missing | 1,042 (2.5) | 16 (1.9) | 1,026 (2.5) |
| House connected to the sewer network: n (%) | | | |
| Yes | 23,608 (56.5) | 477 (57.7) | 23,131 (56.5) |
| No | 15,234 (36.5) | 276 (33.4) | 14,958 (36.5) |
| Missing | 2,920 (7.0) | 73 (8.8) | 2,847 (7.0) |
| Number of residents in the household: n (%) | | | |
| None | 3 (0.0) | - | 3 (0.0) |

(*Continued*)

**Table 3.** (Continued)

| Variables | Total | Positive | Negative |
|---|---|---|---|
| One | 415 (1.0) | 12 (1.5) | 403 (1.0) |
| Two | 10,993 (26.3) | 206 (24.9) | 10,787 (26.4) |
| Three | 11,105 (26.6) | 207 (25.1) | 10,898 (26.6) |
| More than three | 15,404 (36.9) | 330 (40.0) | 15,074 (36.8) |
| Missing | 3,842 (9.2) | 71 (8.6) | 3,771 (9.2) |
| Has fruit trees: n (%) | | | |
| Yes | 7,545 (18.1) | 119 (14.4) | 7,426 (18.1) |
| No | 27,282 (65.3) | 555 (67.2) | 26,727 (65.3) |
| Missing | 6,935 (16.6) | 152 (18.4) | 6,783 (16.6) |
| Has a vegetable garden: n (%) | | | |
| Yes | 3,716 (8.9) | 49 (5.9) | 3,667 (9.0) |
| No | 31,323 (75.0) | 626 (75.8) | 30,697 (75.0) |
| Missing | 6,723 (16.1) | 151 (18.3) | 6,572 (16.1) |
| Family income: n (%) | | | |
| Less than or equal to R$ 500.00 | 16,575 (39.7) | 331 (40.1) | 16,244 (39.7) |
| Between R$ 501.00 and R$ 1000.00 | 10,617 (25.4) | 205 (24.8) | 10,412 (25.4) |
| More than R$ 1001.00 | 3,381 (8.1) | 49 (5.9) | 3,332 (8.1) |
| Missing | 11,189 (26.8) | 241 (29.2) | 10,948 (26.7) |
| Housing status: n (%) | | | |
| Owned | 23,450 (56.2) | 398 (48.2) | 23,052 (56.3) |
| Rented | 8,880 (21.3) | 235 (28.5) | 8,645 (21.1) |
| Donated | 7,101 (17.0) | 137 (16.6) | 6,964 (17.0) |
| Missing | 2,331 (5.6) | 56 (6.8) | 2,275 (5.6) |
| Type of water treatment used: n (%) | | | |
| Filtered | 9,103 (21.8) | 144 (17.4) | 8,959 (21.9) |
| Boiled | 834 (2.0) | 23 (2.8) | 811 (2.0) |
| Disinfected (chlorine) | 22,503 (53.9) | 441 (53.4) | 22,062 (53.9) |
| None | 5,795 (13.9) | 133 (16.1) | 5,662 (13.8) |
| Missing | 3,527 (8.4) | 85 (10.3) | 3,442 (8.4) |
| Age: mean (SD) | 25.2 (4.6) | 24.9 (4.7) | 25.2 (4.6) |

n: number; %: proportional percentage value; SD: standard deviation.

positive cases), we used the random undersampling technique to reduce the difference between the positive and negative congenital syphilis cases, setting a ratio of 55% of the number of samples in the minority class (positive cases) over the number of samples in the majority class (negative cases) after resampling.

2. **Balanced Data Set (BDS)**: balanced data set using the random undersampling technique with 1,652 records (826 positive cases and 826 negative cases).

3. **Imbalanced with One-hot Encoding Data Set (IODS)**: imbalanced data set with one-hot encoding technique applied to transform categorical data into binary data. In this case, the number of attributes increased to 97, since the one-hot encoding creates a new attribute for each class of a given categorical attribute. For example, the categorical attribute that indicates whether the pregnant woman is a smoker (SMOKER) has three possible categories: positive, negative, and not informed. Upon application of the one-hot encoding technique,

this single attribute will be split into three binary attributes: (i) positive for smokers; (ii) negative for smokers; and (iii) not informed.

4. **Balanced with One-hot Encoding Data Set (BODS)**: balanced data set with one-hot encoding technique applied.

5. **Imbalanced with One-hot Encoding with Column Drop Data Set (IODDS)**: imbalanced data set with one-hot encoding technique applied with the column related to not informed by the patient removed from the data set. Some attributes have a class that represents the missing data. In this experiment, after applying the one-hot, we removed the column related to that, decreasing the number of attributes to 75.

6. **Balanced with One-hot Encoding with Column Drop Data Set (BODDS)**: balanced data set and one-hot encoding with the column related to not informed by the patient removed from the data set.

Fig 3 illustrates how the data set was configured for each experiment.

Fig 4 presents the methodology used to train and test our models. After the creation of data sets for each experiment, they were split into training (80%) and testing (20%) sets. In order to evaluate the best approach to selecting the most effective subset of attributes, we applied two different training processes to all experiments, using distinct feature selection techniques (SFA

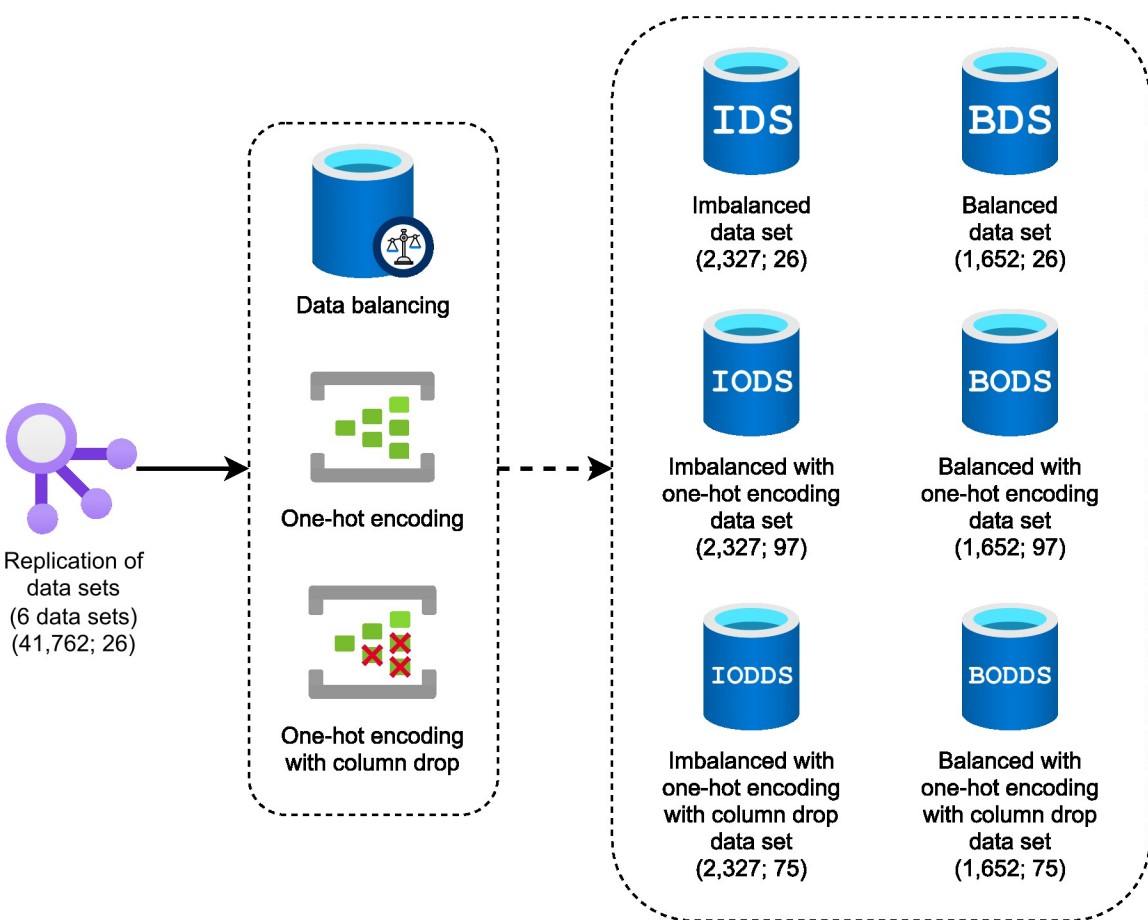

**Fig 3. Creation of data set for the six experiment using data balancing and one-hot encoding techniques.**

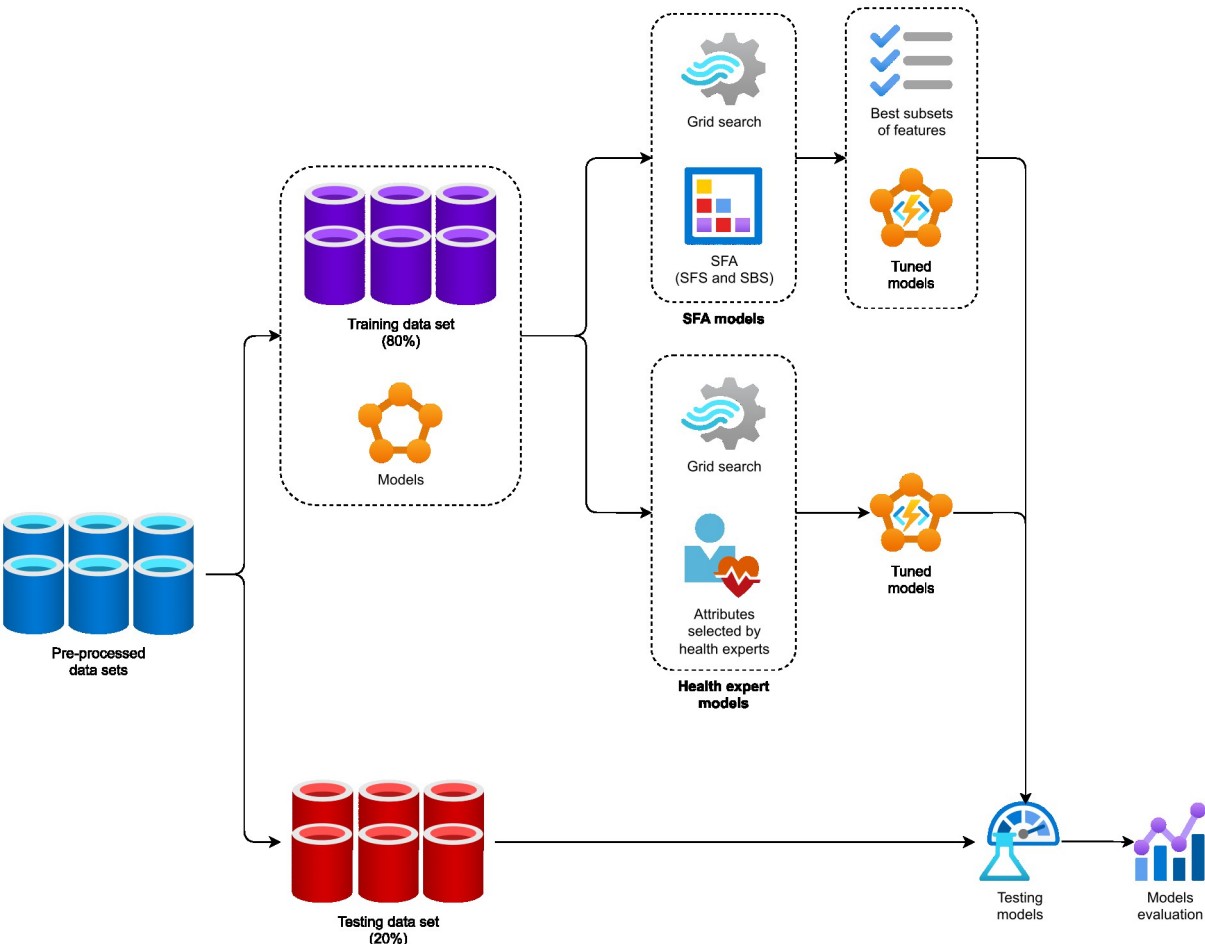

**Fig 4. Design flowchart of models training and testing.**

models and health expert models), resulting in 126 tuned models to be evaluated. We used the hold-out method for models' evaluation.

**SFA models.** In order to automatically find the best subset of attributes (feature selection) and the best model configuration (hyperparameter optimization), in a given search space, two techniques were applied: (*i*) SFA and (*ii*) grid search, respectively. For each combination of the grid search technique (Table 4 presents the hyperparameters used in the grid search), two flavors of the SFA technique were performed ((SFS) and (SBS)).

All executions were based on the accuracy metric with cross-validation ($k = 10$). At the end, the testing data set had its dimensionality reduced based on the best subsets of features for each model, and it was used to evaluate them.

**Health expert models.** As a complement to our experiments, we consulted the health experts from PMCP and asked them to manually select a subset of attributes from the 26 described in Table 2. They analyzed those attributes and selected 11 of them, in addition to the target attribute, highlighted in Table 2 with a star icon (★). Then, the data sets from all experiments (IDS, BDS, IODS, BODS, IODDS, and BODDS) were filtered to reflect the attributes selected by the health experts.

On the basis of the accuracy metric with cross-validation ($k = 10$), the grid search technique was also used to identify the best model configuration in a given search space.

Table 4. Hyperparameters used in the grid search.

| Model | Hyperparameters | Values |
|---|---|---|
| Decision Tree | criterion<br>splitter | ["gini", "entropy"]<br>["best", "random"] |
| Random Forest | n_estimators<br>criterion | [50, 100, 150]<br>["gini", "entropy"] |
| AdaBoost | n_estimators<br>learning_rate | [50, 100, 150]<br>[0.5, 1] |
| GBM | n_estimators<br>learning_rate<br>loss | [50, 100, 150]<br>[0.5, 1]<br>["deviance", "exponential"] |
| XGBoost | learning_rate<br>max_depth | [0.3, 0.5]<br>[5, 10] |
| KNN | n_neighbors<br>p<br>weights | [5, 10, 15]<br>[1, 2]<br>["uniform", "distance"] |
| SVM | kernel<br>gamma | ["rbf", "linear", "poly", "sigmoid"]<br>["scale", "auto"] |

## Results

In this work, we use clinical and sociodemographic data and two training processes (SFA models and health expert models) to evaluate seven machine learning models under six different experiments. We used the F1-Score metric to compare the models with the best performance from each experiment, that is calculated from a harmonic mean between precision and sensitivity, which are two relevant metrics for analysis of health problems. Tables 5 and 6 presents the top-3 best models of each experiment that used attributes selected by the SFA technique and by health experts, respectively. The best model of each experiment is highlighted with an upside down triangle icon (▼) and the best model among the experiments are highlighted with a circle icon (•).

The experiments took more than 5 months to be completed and were performed using four (AWS) Amazon Elastic Compute Cloud (EC2) instances of type c5a.4xlarge with $2^{nd}$ generation AMD EPYC 7002 series processor, 16 vCPUs, 32 GB of RAM, and 30 GB SSD.

The best SFA models of each experiment are compared in Fig 5, where it is possible to notice different results especially for F1-Score, sensitivity, and specificity metrics. These models presented subsets of attributes that ranged between 13 and 42 attributes, and the most common attributes were: EDUC_LEVEL, MARITAL_STATUS, FOOD_INSECURITY, WATER_TREATMENT, and SMOKER, which appeared in almost all experiments.

The model that obtained the highest F1-Score was the SVM when executing the BDS experiment, called SVM-BDS-SFA, using the SBS, with 13 attributes, *gama = scale* and *kernal = rbf*. It reached a F1-Score of 63.04%, an accuracy of 61.03%, a precision of 60.11%, a sensitivity of 66.27%, and a specificity of 55.76%. On the other hand, the Decision Tree in the IODS experiment, with 11 attributes selected by the SBS, *criterion = entropy* and *splitter = best*, had the worst performance among the top-3 of the SFA models, with a F1-Score of 30.53%.

For the health expert models, Fig 6 presents a comparison between the best models of each experiment. The model with the best performance was the AdaBoost when executing the BODS experiment, called AdaBoost-BODS-Expert, with *learning_rate* = 0.5 and *n_estimators* = 150, which achieved a F1-Score of 63.51%, an accuracy of 60.42%, a precision of 59.07%, a sensitivity of 68.67%, and a specificity of 52.12%. Meanwhile, the XGBoost in the IODDS

**Table 5. Results of the three best models of each experiment that used the SFA technique to select features.**

| Model | SFA | Qty. Att. | F1-Score | Accuracy | Precision | Sensitivity | Specificity |
|---|---|---|---|---|---|---|---|
| **IDS** | | | | | | | |
| Random Forest▼ | SBS | 15 | 34.85% | 63.09% | 47.42% | 27.54% | 82.94% |
| KNN | SBS | 16 | 33.71% | 62.45% | 45.83% | 26.35% | 82.61% |
| AdaBoost | SBS | 10 | 30.71% | 64.16% | 50.00% | 22.16% | 87.63% |
| **BDS**▼ | | | | | | | |
| SVM ▼ · | SBS | 13 | 63.04% | 61.03% | 60.11% | 66.27% | 55.76% |
| AdaBoost | SFS | 9 | 61.33% | 57.70% | 56.63% | 66.87% | 48.48% |
| GBM | SBS | 12 | 58.08% | 57.70% | 57.74% | 58.43% | 56.97% |
| **IODS**▼ | | | | | | | |
| Random Forest▼ | SBS | 42 | 35.52% | 64.16% | 50.00% | 27.54% | 84.62% |
| KNN | SBS | 34 | 31.91% | 58.80% | 39.13% | 26.95% | 76.59% |
| Decision Tree | SBS | 11 | 30.53% | 60.94% | 42.11% | 23.95% | 81.61% |
| **BODS**▼ | | | | | | | |
| Decision Tree▼ | SBS | 16 | 60.44% | 56.50% | 55.56% | 66.27% | 46.67% |
| SVM | SBS | 37 | 60.41% | 59.21% | 58.86% | 62.05% | 56.36% |
| GBM | SFS | 32 | 59.71% | 58.01% | 57.54% | 62.05% | 53.94% |
| **IODDS**▼ | | | | | | | |
| XGBoost▼ | SBS | 16 | 43.92% | 64.38% | 50.39% | 38.92% | 78.60% |
| KNN | SBS | 41 | 37.80% | 61.16% | 44.35% | 32.93% | 76.92% |
| Random Forest | SBS | 39 | 35.48% | 65.67% | 54.32% | 26.35% | 87.63% |
| **BODDS**▼ | | | | | | | |
| SVM▼ | SFS | 14 | 59.08% | 59.82% | 60.38% | 57.83% | 61.82% |
| AdaBoost | SFS | 26 | 58.38% | 56.50% | 56.11% | 60.84% | 52.12% |
| Decision Tree | SFS | 11 | 58.14% | 56.50% | 56.18% | 60.24% | 52.73% |

▼ Best model of the experiment;

· Best model among the experiments.

experiment reached a F1-Score of 34.11%, the worst one among the top-3 of the health experts experiments.

The SVM-BDS-SFA and the AdaBoost-BODS-Expert models achieved similar results, presenting major differences in the sensitivity, specificity, and attributes used. Fig 7 presents a comparison between them, and Table 7 presents the grid search results and the attributes selected for both models.

## Discussion

The results of the experiments in Tables 5 and 6 presented a large variation of F1-Score values, ranging from 30.71% to 63.51%. In general, the experiments with imbalanced data set, which had a greater number of negative cases than positive cases of congenital syphilis, obtained inferior results than their respective experiment with balanced data set, creating models with a low rate of correct classification of truly positive cases (low sensitivity), directly affecting the F1-Score, and with a high rate of correct classification of truly negative cases (high specificity). Therefore, such models are not relevant to the purpose of this work, since we aim to predict positive cases of congenital syphilis.

It was possible to observe that the experiments that used the one-hot encoding technique, dropping or not dropping the attribute related to not informed data, produced results that

**Table 6. Results of the three best models of each experiment that used the attributes selected manually by health experts.**

| Model | F1-Score | Accuracy | Precision | Sensitivity | Specificity |
|---|---|---|---|---|---|
| **IDS** | | | | | |
| Random Forest▼ | 43.83% | 60.94% | 45.22% | 42.51% | 71.24% |
| Decision Tree | 38.41% | 56.65% | 39.13% | 37.72% | 67.22% |
| GBM | 36.90% | 63.30% | 48.08% | 29.94% | 81.94% |
| **BDS** | | | | | |
| KNN▼ | 62.37% | 57.70% | 56.31% | 69.88% | 45.45% |
| AdaBoost | 62.01% | 58.91% | 57.81% | 66.87% | 50.91% |
| Random Forest | 61.99% | 60.73% | 60.23% | 63.86% | 57.58% |
| **IODS** | | | | | |
| Decision Tree▼ | 44.84% | 59.87% | 44.19% | 45.51% | 67.89% |
| GBM | 35.84% | 61.59% | 44.64% | 29.94% | 79.26% |
| XGBoost | 34.11% | 57.73% | 38.64% | 30.54% | 72.91% |
| **BODS** | | | | | |
| AdaBoost▼· | 63.51% | 60.42% | 59.07% | 68.67% | 52.12% |
| SVM | 62.98% | 59.52% | 58.16% | 68.67% | 50.30% |
| GBM | 59.39% | 59.52% | 59.76% | 59.04% | 60.00% |
| **IODDS** | | | | | |
| Decision Tree▼ | 41.79% | 58.15% | 41.67% | 41.92% | 67.22% |
| Random Forest | 38.41% | 63.52% | 48.62% | 31.74% | 81.27% |
| XGBoost | 36.84% | 58.80% | 40.88% | 33.53% | 72.91% |
| **BODDS** | | | | | |
| SVM▼ | 62.01% | 58.91% | 57.81% | 66.87% | 50.91% |
| AdaBoost | 61.80% | 58.91% | 57.89% | 66.27% | 51.52% |
| Random Forest | 60.66% | 60.42% | 60.48% | 60.84% | 60.00% |

▼ Best model of the experiment;

· Best model among the experiments.

were not significantly different from those that did not use it. The large amount of missing data may have impacted the results of these models, impairing the models' learning.

For the SFA models, only experiments with imbalanced data set benefited from the one-hot encoding technique. When we compared the best models from the IODS and IODDS experiments with the best model from the IDS experiment, we found a slight improvement in almost all metrics, particularly for the XGBoost model from the IODDS experiment, which benefited from the dropping of the attribute related to not informed data, obtaining a higher sensitivity (38.92%), resulting in a better F1-Score value of 43.92%. Nonetheless, the XGBoost model did not achieve relevant results.

The one-hot encoding technique improved all health expert models. For experiments with imbalanced data set, the F1-Score value of the Decision Tree model from the IODS experiment (44.84%) was slightly higher than the Random Forest model from the IDS experiment (43.83%). The Decision Tree model from the IODDS experiment, on the other hand, obtained a lower F1-Score value (41.79%), indicating that there was no benefit in dropping the attribute related to not informed data. As for the experiments with balanced data set, the AdaBoost model from the BODS experiment (AdaBoost-BODS-Expert) and the SVM model from the BODDS experiment performed a little better than the KNN from the BDS experiment, except for sensitivity.

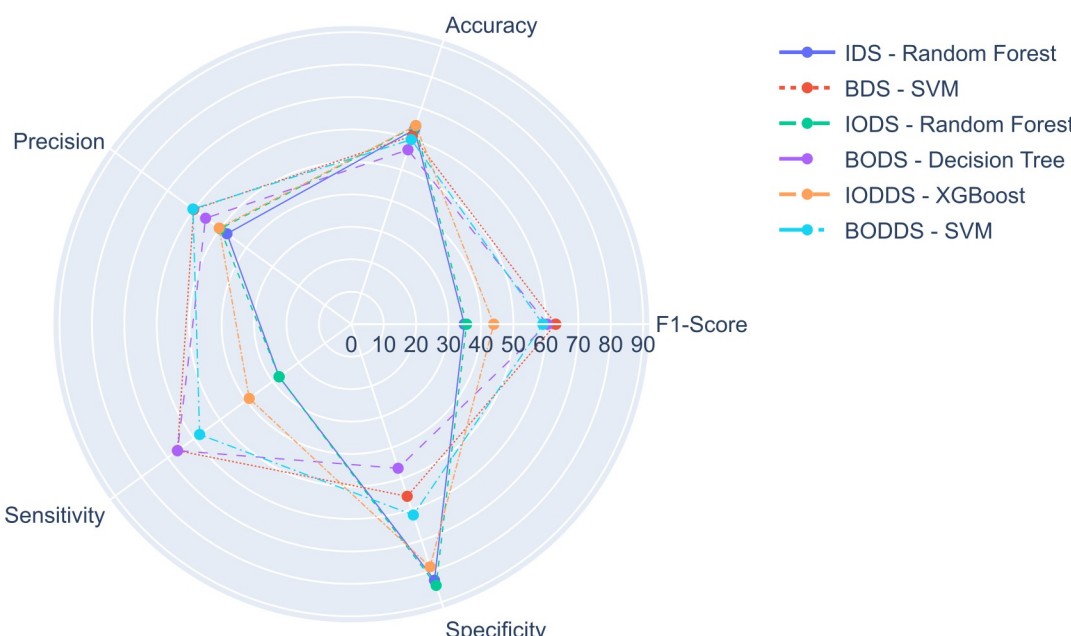

**Fig 5. Comparison of the best SFA models among all experiments.**

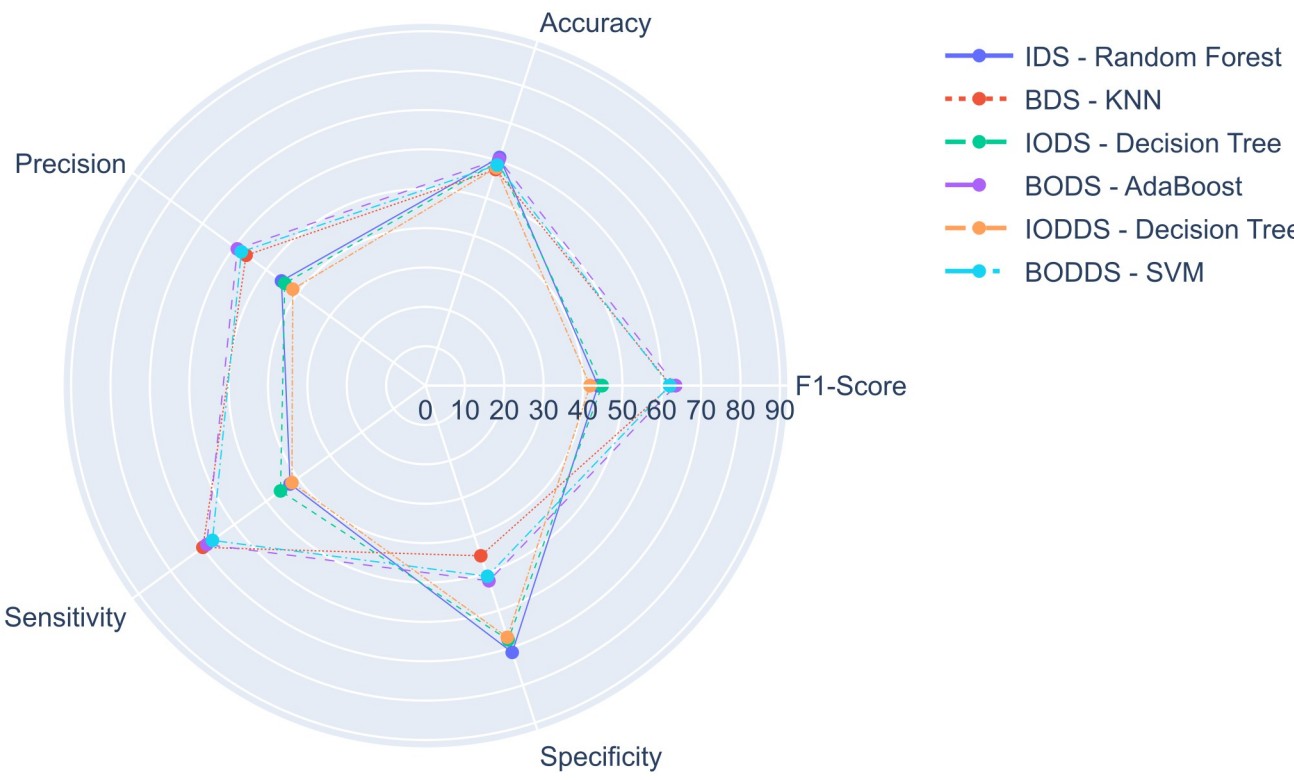

**Fig 6. Comparison of the best health expert models.**

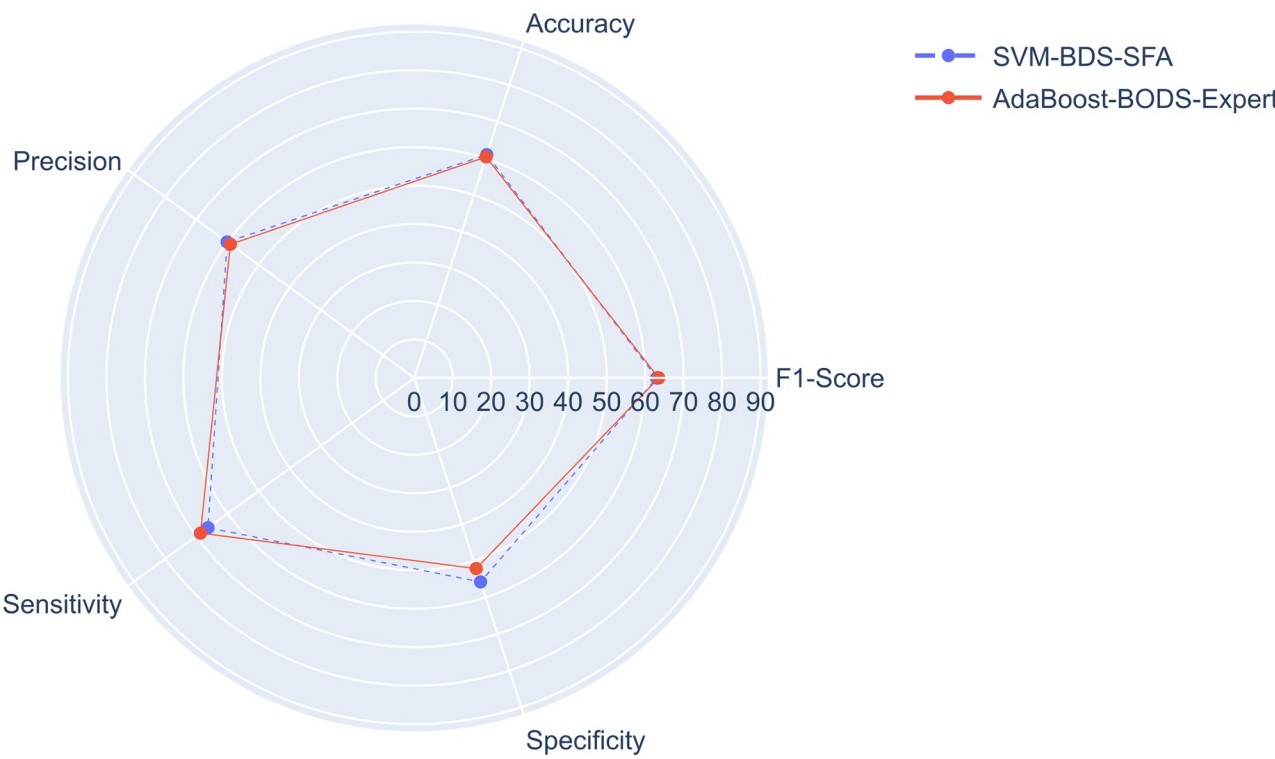

**Fig 7. SVM-BDS-SFA and AdaBoost-BODS-Expert comparision.**

The most effective results, among the models with the best performance from each experiment, were provided by the tree-based models Random Forest, Decision Tree, AdaBoost, and XGBoost, standing out in four out of six experiments, both with attributes chosen through SFA technique and those chosen by health experts. Tree-based models are typically good choices for problems involving tabular data [42].

The best results were obtained by the SVM-BDS-SFA and AdaBoost-BODS-Expert models, which produced fairly similar results. Seven of the 11 attributes selected by health experts used by the AdaBoost-BODS-Expert model were also chosen for the SVM-BDS-SFA via the SFA technique, demonstrating that more than half of the attributes chosen by health experts are still relevant even when using SFA technique.

**Table 7. Grid search and feature selection results.**

| Model | Hyperparameters | Qty. Att. | Attributes |
|---|---|---|---|
| SVM-BDS-SFA | gamma: scale<br>kernel: rbf | 13 | EDUC_LEVEL, HAS_FRU_TREE, WATER_TREATMENT, RH_FACTOR, PLAN_PREGNANCY, HAS_PREG_RISK, TET_VACCINE, IS_HEAD_FAMILY, MARITAL_STATUS, FOOD_INSECURITY, NUM_ABORTIONS, NUM_PREGNANCIES, and HAS_FAM_INCOME |
| AdaBoost-BODS-Expert | learning_rate: 0.5<br>n_estimators: 150 | 11 | AGE, LEVEL_SCHOOLING, FAM_INCOME, PLAN_PREGNANCY, HAS_PREG_RISK, MARITAL_STATUS, FOOD_INSECURITY, NUM_ABORTIONS, NUM_LIV_CHILDREN, NUM_PREGNANCIES, and FAM_PLANNING |

**SVM-BDS-SFA**: SVM model that used the SFA technique to select attributes; **AdaBoost-BODS-Expert**: AdaBoost model that used attributes selected manually by health experts.

The AdaBoost-BODS-Expert model is more interpretable by health experts and, as a result, would be more favored for usage by PMCP, despite having a slightly lower performance when compared to the SVM-BDS-SFA model. In this sense, the AdaBoost-BODS-Expert model has a significant advantage over the SVM-BDS-SFA model, with approximately 2% difference in results for each metric. Even if it does not produce the greatest outcomes, a model that provides interpretability and knowledge of all attributes used might enable higher acceptance by health experts, producing more confidence and allowing adoption in daily usage [43].

None of the models presented an accuracy greater than 70%, ranging from 56.50% to 64.38%, demonstrating the difficulty of classifying possible outcomes of congenital syphilis using only clinical and sociodemographic data. The reason that may explain this fact is the abundance of missing data that reduces the data quality and model learning [44]. In order to cover this issue, we created a categorical value that represented that one that was not informed by the patient. However, this may have caused: (*i*) difficulty of finding patterns in the data that would allow a more accurate classifications and (*ii*) relevance for categories related to not informed data, which may hinder the classification of more realistic data, when the attributes are correctly filled. As potential improvements, we recommend (*i*) the use of data imputation technique to reduce the proportion of missing data in the existing data set, as well as (*ii*) a better completeness of the data by the PMCP, which is a significant improvement that allows the data to more precisely reflect reality.

## Conclusions and future works

Congenital syphilis has become a serious public health problem, with a significant increase in the number of cases in Brazil [7]. The PHC has tests and treatments for congenital syphilis, but the cost of these tests and the fact that pregnant women don't always follow the treatments have made it a significant issue. In this work, machine learning models were evaluated for classification of possible congenital syphilis outcomes in pregnancies assisted by the PMCP.

We evaluated seven machine learning techniques for the prediction of congenital syphilis cases: Decision Tree, Random Forest, AdaBoost, GBM, XGBoost, KNN, and SVM. We proposed six experiments (IDS, BDS, IODS, BODS, IODDS, and BODDS), executing a grid search for each model along with the SFA technique (SFA models). We also executed all experiments only with the grid search, using 11 attributes selected by health experts from PMCP (health expert models).

The SVM model from the BDS experiment that used the SFA technique, called SVM-BDS-SFA, and the AdaBoost model from the BODS experiment with attributes chosen by health experts, named AdaBoost-BODS-Expert, produced the best results in terms of F1-Score metric.

When comparing metrics of SVM-BDS-SFA and AdaBoost-BODS-Expert models, both models obtained similar outcomes, although the SVM-BDS-SFA model outperformed the AdaBoost-BODS-Expert model in almost all metrics, except sensitivity and F1-Score.

Our results showed that it is possible to predict congenital syphilis during pregnancy, only using clinical and sociodemographic data. At the same time, we also identified that the large amount of missing data may have precluded more accurate results, indicating the need for the PMCP to improve data acquisition quality.

As future work, we plan to expand our work to predict gestational syphilis. The timely identification of syphilis cases in pregnant women can help to reduce the incidence of congenital syphilis. We also plan to apply different techniques to handle the missing data, and retrain the proposed models applying a statistical analysis to evaluate the results.

## Acronyms

AAPC: Average Annual Percent Change AdaBoost: Adaptive Boosting AUC ROC: Area Under The Receiver Operating Characteristics Curve AWS: Amazon Web Services BDS: Balanced Data Set BODDS: Balanced with One-hot Encoding with Column Drop Data Set BODS: Balanced with One-hot Encoding Data Set CAAE: Certificate of Presentation of Ethical Appreciation CATWOE: Customers, Actors, Transformation process, Weltanschauung, Ownership, and Environmental constrains CEP: *Comitê de Ética em Pesquisa* EC2: Amazon Elastic Compute Cloud FN: False Negative FP: False Positive GBM: Gradient Boosting Machines GUI: Graphical User Interface HDI: Human Development Index HIV: Human Immunodeficiency Virus IBGE: *Instituto Brasileiro de Geografia e Estatística* ICD-10: International Classification of Diseases 10th Revision IDS: Imbalanced Data Set IODDS: Imbalanced with One-hot Encoding with Column Drop Data Set IODS: Imbalanced with One-hot Encoding Data Set KNN: K-Nearest Neighbors OAS: Organization of American States PAHO: Pan American Health Organization PHC: Primary Health Care PMAQ: National Program for Improvement of Access and Quality of Primary Care PMCP: *Mãe Coruja Pernambucana* Program RF: Random Forest SBS: Sequential Backward Selection SFA: Sequential Feature Algorithm SFS: Sequential Forward Selection SIH: *Sistema de Informações Hospitalares* SIM: *Sistema de Informação Sobre Mortalidade* SINAN: *Sistema de Informação de Agravos de Notificação* SINASC: *Sistema de Informações sobre Nascidos Vivos* SIS-MC: *Sistema de Informação do Mãe Coruja* STD: Sexually Transmitted Diseases STIs: Sexually Transmitted Infections SUS: *Sistema Único de Saúde* SVM: Support Vector Machine TN: True Negative TP: True Positive UFPE: *Universidade Federal de Pernambuco* UN: United Nations UNAIDS: Joint United Nations Program on HIV/AIDS VDRL: Venereal Disease Research Laboratory WHO: World Health Organization XGBoost: eXtreme Gradient Boosting.

## Acknowledgments

Authors would like to thank Conselho Nacional de Desenvolvimento Científico e Tecnológico (CNPq); Fundação de Amparo a Ciência e Tecnologia do Estado de Pernambuco (FACEPE); Programa Mãe Coruja Pernambucana, Secretaria de Saúde do Estado de Pernambuco; Universidade Federal de Pernambuco (UFPE), Universidade de Pernambuco (UPE), an entity of the Government of the State of Pernambuco focused on the promotion of Teaching, Research and Extension; and Universidade Federal da Paraíba (UFPB).

## Author Contributions

**Conceptualization:** Igor Vitor Teixeira, Patricia Takako Endo.

**Data curation:** Igor Vitor Teixeira, Morgana Thalita da Silva Leite, Flávio Leandro de Morais Melo, Élisson da Silva Rocha, Patricia Takako Endo.

**Formal analysis:** Igor Vitor Teixeira, Morgana Thalita da Silva Leite, Flávio Leandro de Morais Melo, Élisson da Silva Rocha, Patricia Takako Endo.

**Funding acquisition:** Cleber Matos de Morais, Judith Kelner.

**Investigation:** Igor Vitor Teixeira, Patricia Takako Endo.

**Methodology:** Igor Vitor Teixeira, Patricia Takako Endo.

**Project administration:** Igor Vitor Teixeira, Patricia Takako Endo.

**Supervision:** Cleber Matos de Morais, Judith Kelner, Patricia Takako Endo.

**Validation:** Igor Vitor Teixeira, Ana Sofia Pessoa da Costa Carrarine, Marília Santana, Cristina Pinheiro Rodrigues, Ana Maria de Lima Oliveira, Keduly Vieira Gadelha, Patricia Takako Endo.

**Visualization:** Patricia Takako Endo.

**Writing – original draft:** Igor Vitor Teixeira, Morgana Thalita da Silva Leite, Flávio Leandro de Morais Melo, Élisson da Silva Rocha, Sara Sadok, Cleber Matos de Morais, Patricia Takako Endo.

**Writing – review & editing:** Igor Vitor Teixeira, Élisson da Silva Rocha, Sara Sadok, Ana Sofia Pessoa da Costa Carrarine, Marília Santana, Cristina Pinheiro Rodrigues, Ana Maria de Lima Oliveira, Keduly Vieira Gadelha, Cleber Matos de Morais, Judith Kelner, Patricia Takako Endo.

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
