## [Decision Letter · Decision Letter 0]

18 Nov 2022

PONE-D-22-27034Predicting congenital syphilis cases: a performance evaluation of different machine learning modelsPLOS ONE

Dear Dr. Endo,

Thank you for submitting your manuscript to PLOS ONE. After careful consideration, we feel that it has merit but does not fully meet PLOS ONE’s publication criteria as it currently stands. Therefore, we invite you to submit a revised version of the manuscript that addresses the points raised during the review process.

We look forward to receiving your revised manuscript.

Kind regards,

Anwar P.P. Abdul Majeed

Academic Editor

PLOS ONE

Journal Requirements:

2. Please note that PLOS ONE has specific guidelines on code sharing for submissions in which author-generated code underpins the findings in the manuscript. In these cases, all author-generated code must be made available without restrictions upon publication of the work. 

Please review our guidelines at https://journals.plos.org/plosone/s/materials-and-software-sharing#loc-sharing-code and ensure that your code is shared in a way that follows best practice and facilitates reproducibility and reuse. New software must comply with the Open Source Definition.

"This work was partially funded by Conselho Nacional de Desenvolvimento Cient´ıfico e Tecnol´ogico (CNPq), Coordena¸c˜ao de Aperfei¸coamento de Pessoal de N´ıvel Superior (CAPES), Funda¸c˜ao de Amparo a Ciˆencia e Tecnologia do Estado de Pernambuco (FACEPE), and Universidade de Pernambuco (UPE), an entity of the Government of the State of Pernambuco focused on the promotion of teaching, research and extension."

"JK

OPP1202194

Bill & Melinda Gates Foundation

https://www.gatesfoundation.org/

"JK

OPP1202194

Bill & Melinda Gates Foundation

https://www.gatesfoundation.org/

Reviewers' comments:

Reviewer's Responses to Questions

**Comments to the Author**

1. Is the manuscript technically sound, and do the data support the conclusions?

Reviewer #1: Yes

2. Has the statistical analysis been performed appropriately and rigorously? 

Reviewer #1: No

3. Have the authors made all data underlying the findings in their manuscript fully available?

Reviewer #1: Yes

4. Is the manuscript presented in an intelligible fashion and written in standard English?

Reviewer #1: Yes

5. Review Comments to the Author

Reviewer #1: The paper was written fairly well, however several issues need to be addressed to consider as a good technical paper.

Abstract

The number of dataset is not defined here as well as the technique of feature selection.

The results defined was in general, without including statistical results.

Result and Discussion

The result obtain from the study is too low compared to reference [7].

6. PLOS authors have the option to publish the peer review history of their article (what does this mean?). If published, this will include your full peer review and any attached files.

Reviewer #1: No

---

## [Author Response · Author response to Decision Letter 0]

5 Dec 2022

Reviewer #1: The paper was written fairly well, however several issues need to be addressed to consider as a good technical paper.

Dear reviewer, we do appreciate your comments and suggestions to improve our work. We have addressed them properly.

Abstract

The number of dataset is not defined here as well as the technique of feature selection.

In Section Data set, we described the number of data set and tables as follow: 

"We used nine tables with anonymized data provided by the PMCP, extracted from their information system, named SIS-MC. These tables contain clinical and sociodemographic data regarding antenatal care, pregnant women's outcomes, and their children, from the cities served by the PMCP in the State of Pernambuco, Brazil, between the years of 2013 and 2021 [...]".

"Fig 1 illustrates the data pre-processing methodology designed to unify the nine tables provided by the PMCP into an unified dataset [...]".

In the Section Data set, we updated Fig 1 to illustrate the unification of the multiple tables into a unique data set, called a unified data set. We also updated the URL of the pre-processed data set publicly available at Data Mendeley.

In Section SFA models, we said: 

"In order to automatically find the best subset of attributes (feature selection) and the best model configuration (hyperparameter optimization), in a given search space, two techniques were applied: (i) SFA and(ii) grid search, respectively. For each combination of the grid search technique (Table 3 presents the hyperparameters used in the grid search), two flavors of the SFA technique were performed (SFS and SBS)." 

In Section Hyper-parameter optimization and feature selection techniques, we have described both techniques:

"The SFS flavor starts with an empty subset of attributes; with each iteration, a new attribute is added, thereby selecting the attributes that increase the model's performance. On the other hand, the SBS flavor begins with a subset including all the attributes of the data set; with each iteration, different combinations of attributes are evaluated, and the attribute with the least impact on model performance is removed."

The results defined was in general, without including statistical results.

As we did not make repetitions in our experiments, we can not include statistical results in this current work. In total, we had 126 models that were trained using AWS resources and took too long to be finished (more than 5 months). Therefore, due to the time needed to retrain the models and understanding the relevance of this question, we have added statistical analysis as future work.

Result and Discussion

The result obtain from the study is too low compared to reference [7].

We did not understand this point. Our reference [7] is an nationwide epidemiological bulletin from the Brazilian Ministry of Health. We are not able to compare results, as our dataset is restricted to specific samples in only one state (Pernambuco) and [7] does not have prediction results.

---

## [Decision Letter · Decision Letter 1]

5 Feb 2023

PONE-D-22-27034R1Predicting congenital syphilis cases: a performance evaluation of different machine learning modelsPLOS ONE

Dear Dr. Endo,

Thank you for submitting your manuscript to PLOS ONE. After careful consideration, we feel that it has merit but does not fully meet PLOS ONE’s publication criteria as it currently stands. Therefore, we invite you to submit a revised version of the manuscript that addresses the points raised during the review process.

We look forward to receiving your revised manuscript.

Kind regards,

Anwar P.P. Abdul Majeed

Academic Editor

PLOS ONE

Journal Requirements:

Reviewers' comments:

Reviewer's Responses to Questions

**Comments to the Author**

1. If the authors have adequately addressed your comments raised in a previous round of review and you feel that this manuscript is now acceptable for publication, you may indicate that here to bypass the “Comments to the Author” section, enter your conflict of interest statement in the “Confidential to Editor” section, and submit your "Accept" recommendation.

Reviewer #1: (No Response)

2. Is the manuscript technically sound, and do the data support the conclusions?

Reviewer #1: Yes

3. Has the statistical analysis been performed appropriately and rigorously? 

Reviewer #1: N/A

4. Have the authors made all data underlying the findings in their manuscript fully available?

Reviewer #1: Yes

5. Is the manuscript presented in an intelligible fashion and written in standard English?

Reviewer #1: Yes

6. Review Comments to the Author

Reviewer #1: The paper was written fairly well, however there is some issue need to be addressed to consider as a good technical paper.

The author state that table 3 in refer to "the hyperparameters used in the grid search", however in the manuscript table 3 caption is "Baseline characteristics of the data set.". Please make an amendment for this table caption

7. PLOS authors have the option to publish the peer review history of their article (what does this mean?). If published, this will include your full peer review and any attached files.

Reviewer #1: No

---

## [Author Response · Author response to Decision Letter 1]

14 Feb 2023

Reviewer #1: The author state that table 3 in refer to "the hyperparameters used in the grid search", however in the manuscript table 3 caption is "Baseline characteristics of the data set.". Please make an amendment for this table caption

Dear reviewer, we do appreciate your comment and suggestions to improve our work. 

The overall baseline characteristics of the pre-processed data set related to the pregnant women assisted by the PMCP are presented in Table 3; and Table 4 presents the hyperparameters used in the grid search.

---

## [Decision Letter · Decision Letter 2]

28 Mar 2023

Predicting congenital syphilis cases: a performance evaluation of different machine learning models

PONE-D-22-27034R2

Dear Dr. Endo,

We’re pleased to inform you that your manuscript has been judged scientifically suitable for publication and will be formally accepted for publication once it meets all outstanding technical requirements.

Kind regards,

Anwar P.P. Abdul Majeed

Academic Editor

PLOS ONE

Additional Editor Comments (optional):

Reviewers' comments:

Reviewer's Responses to Questions

**Comments to the Author**

1. If the authors have adequately addressed your comments raised in a previous round of review and you feel that this manuscript is now acceptable for publication, you may indicate that here to bypass the “Comments to the Author” section, enter your conflict of interest statement in the “Confidential to Editor” section, and submit your "Accept" recommendation.

Reviewer #1: All comments have been addressed

2. Is the manuscript technically sound, and do the data support the conclusions?

Reviewer #1: Yes

3. Has the statistical analysis been performed appropriately and rigorously? 

Reviewer #1: No

4. Have the authors made all data underlying the findings in their manuscript fully available?

Reviewer #1: Yes

5. Is the manuscript presented in an intelligible fashion and written in standard English?

Reviewer #1: Yes

6. Review Comments to the Author

Reviewer #1: (No Response)

7. PLOS authors have the option to publish the peer review history of their article (what does this mean?). If published, this will include your full peer review and any attached files.

Reviewer #1: No

---

## [Editor Report · Acceptance letter]

22 May 2023

PONE-D-22-27034R2 

Predicting congenital syphilis cases: a performance evaluation of different machine learning models  

Dear Dr. Endo:

I'm pleased to inform you that your manuscript has been deemed suitable for publication in PLOS ONE. Congratulations! Your manuscript is now with our production department. 

Kind regards, 

on behalf of

Dr. Anwar P.P. Abdul Majeed 

Academic Editor

PLOS ONE